# Does Hamiltonian Replica Exchange via Lambda-Hopping Enhance the Sampling in Alchemical Free Energy Calculations?

**DOI:** 10.3390/molecules27144426

**Published:** 2022-07-11

**Authors:** Piero Procacci

**Affiliations:** Chemistry Department, University of Florence, Via Lastruccia n.3, I-50019 Sesto Firentino, Italy; piero.procacci@unifi.it

**Keywords:** drug design, molecular dynamics, binding free energy, replica exchange, FEP, FEP^+^, solute tempering

## Abstract

In the context of computational drug design, we examine the effectiveness of the enhanced sampling techniques in state-of-the-art free energy calculations based on alchemical molecular dynamics simulations. In a paradigmatic molecule with competition between conformationally restrained E and Z isomers whose probability ratio is strongly affected by the coupling with the environment, we compare the so-called λ-hopping technique to the Hamiltonian replica exchange methods assessing their convergence behavior as a function of the enhanced sampling protocols (number of replicas, scaling factors, simulation times). We found that the pure λ-hopping, commonly used in solvation and binding free energy calculations via alchemical free energy perturbation techniques, is ineffective in enhancing the sampling of the isomeric states, exhibiting a pathological dependence on the initial conditions. Correct sampling can be restored in λ-hopping simulation by the addition of a “hot-zone” scaling factor to the λ-stratification (FEP^+^ approach), provided that the additive hot-zone scaling factors are tuned and optimized using preliminary ordinary replica-exchange simulation of the end-states.

## 1. Introduction

Modern computational drug design has recently begun raising enormous interest in industrial settings. According to P&S intelligence, the “in-silico drug discovery market is set to reach ≃6 million by 2030” [1]. The in silico-pipeline for drug discovery relies on a preliminary profiling of millions compounds typically via docking in combination with knowledge-based and machine learning techniques, followed by a refinement step (hit-to-lead) using simulations based on an accurate physical description of the system, aimed at eliminating costly false positives and at eventually prioritizing the leads for further wet-lab validation. In this context, the calculation of binding free energy (BFE) in ligand-receptor systems via atomistic molecular dynamics (MD) simulation is one of the major challenge in today’s computational chemistry.

The consensus approach for computing BFEs is based on the so-called alchemical protocol [2], whereby the end-states of the L + R = LR reaction are connected via intermediate nonphysical states in which the ligand(L)-environment interactions are progressively decoupled. The binding free energy is recovered via a thermodynamic cycle as the difference between two solvation free energies, namely that of the ligand in the bound state and in the pure solvent. The intermediate non-physical states (typically few tens) consist in the so-called λ-stratification [3], with λ being the alchemical parameter controlling the level of interaction of the ligand with the environment. The free energy difference between two consecutive λ states is generally computed via free energy perturbation (FEP) [4] or, equivalently, thermodynamic integration (TI) [5]. Notably, these λ-simulations are completely independent and can be performed in parallel, delivering the result in the wall-clock time of a single λ-simulation. On the other hand, as each of the non-physical states along the stratification must be canonically sampled, convergence constitutes the major challenge in alchemical BFE calculations. Incomplete sampling, due to the existence of large free energy barriers between conformational states in ligand and proteins, may produce unreliable results that are strongly dependent on the initial conditions [6,7]. This pathology affects also relative binding free energy (RBFE) calculations involving the transmutation of a ligand into a congeneric compound in both arms of the alchemical thermodynamic cycle [8].

To overcome these serious drawbacks, the alchemical technology is combined with enhanced sampling techniques, typically based on the Hamiltonian Replica Exchange Method (HREM) [9]. In HREM, several MD simulations are launched in parallel with global or local heating with scaled temperature ranging from the desired temperature (the *target* thermodynamic state) up to a high temperature where free energy barriers can be easily overcome. At regular intervals, exchange of coordinates (or equivalently of temperatures) are attempted between contiguous replica with an acceptance probability regulated by a Metropolis criterion. In this fashion, each replica walker spans the entire thermodynamic range of temperatures, transmitting to the target state, with the correct Boltzmann weight, conformations that are sampled in the high-temperature replicas.

The most straightforward HREM implementation of the FEP-based alchemical calculations allows Metropolis-regulated exchanges between λ-states, in the hope that the mixing of states with disparate alchemical coupling could facilitate the sampling [10,11,12]. The current state-of-the art of HREM-based BFE calculations is the so called FEP${}^{+}$ method [13,14]. Here, λ-hopping occurs along a stratification where intra and intermolecular potential of an “hot-region” [13] (typically involving the ligand and nearby residues in the bound state) are scaled with a local temperature reaching the climax at the center of the stratification and being normal at the end-states. FEP${}^{+}$ is believed to efficiently cope with the sampling problem without resorting to expensive 2D HREM simulation schemes on λ and on, e.g., selected torsional degrees of freedom [15].

Despite its widespread use, however, the effectiveness of HREM techniques with λ-hopping or FEP${}^{+}$ in BFE or RBFE calculations has been recently questioned by several authors. According to Baumann et al. [16], “the enhanced sampling technique HREX was not able to overcome inadequate sampling” in ligand–protein systems. In Ref. [17], convergence in HREM-based approaches was found sluggish even in simple host–guest systems [18], requiring a “significant number of iterations”. A similar outcome was found in the 2022 study by Markthaler et al. [19], where rather long (more than 100 ns) HREM simulations were required to “remove an observed starting structure dependence to an acceptable results”. In Ref. [20], in the context of BFE calculations, the authors concluded that “sampling enhancement by means of HREX does not necessarily improve the accuracy of the estimated free energies”. Coveney and coworkers, finally, in their extensive study on RBFE in ligand–protein systems, found “no benefits accrue from replica-exchange methods”, openly questioning the effectiveness of HREM-based methodologies in RBFE calculation, including FEP${}^{+}$ [21].

Here, to test the effectiveness of the HREM-FEP technology, we use as a paradigmatic example the E-Z equilibrium in 5-Aminopent-3-enoic-acid (APA), a zwitterionic molecule in solution characterized by torsional barrier around the central sp2 bond of tens of kcal/mol. We show that while ordinary HREM simulation with solute tempering (ST) [22,23] at full coupling is able to rapidly attain, in few ns, the E-Z conformational equilibrium of a solvated APA molecule at ordinary temperature, HREX-FEP with simple lambda hopping is pathologically dependent on the initial configuration and is ineffective in surmounting the the torsional barrier at any λ value. Paying a significant computational cost, results do improve with FEP${}^{+}$, provided that the additional torsional scaling is brought to the level of that used in a standard HREM.

## 2. Materials and Methods

**APA structure and potential:** In Figure 1, we show the 2D and 3D structures of E-(trans) and Z-(cis) 5-Aminopent-3-enoic acid (APA) diastereoisomers. In water solution at pH = 7, APA is in the zwitterionic state with pKa’s of the carboxylic and amino groups of 4.49 and 9.65, respectively [24]. APA can be selectively obtained in the conformationally restricted Z or E forms. The Z isomer serves as the building block for the synthesis of macrocyclic pseudopeptides for the treatment of a variety of diseases [25].

The E-form is used to synthesize cyclic macrolactames aggregating in remarkable supramolecular structures with potential application in drug delivery, photonics, material science and catalysis [26]. While the E-Z thermodynamic equilibrium could be in principle easily achieved by photoisomerization of the double bond via excitation to the singlet $\pi^{*}$ level [27,28], to our knowledge, no experimentally determined equilibrium cis–trans ratio is available in the literature.

The potential parameters for zwitterionic APA were obtained using the PrimaDORAC web interface [29] based on the well-established GAFF2 force field for organic molecules [30,31]. The barrier height for the torsional potential around the double bond is of the order of 40 kcal/mol, consistently with the experimental indications on non-conjugated olephins [28].

Simulations in the gas phase were done on a single zwitterionic molecule in a large box under constant temperature (300 K) via a Nosé–Hoover thermostat [32]. Simulations in the solution were performed dissolving a single zwitterionic APA molecule in 512 solvent molecules using the OPC3 three-site model for water [33] in conditions of constant pressure (0.1 Mpa) and temperature (300 K), imposed using an NPT-extended Lagrangian with isotropic stress tensor [34]. Electrostatic interactions were treated using the Particle Mesh Ewald technique [35]. All enhanced sampling simulations were performed with the public domain ORAC program [36]. The potential parameters along with all the ORAC input files used in this study can be found on the Zenodo public repository at the link https://zenodo.org/record/6665606 (accessed on 19 June 2022).

**Gas-phase HREM:** For the HREM simulations in the gas phase, we used a minimum scaling factors for the potential energy of 0.1 and 0.05, corresponding to temperatures of 3000 K and 6000 K. The kinetic energy of the APA molecule (Z or E) is unscaled and kept at the normal level of 300 K, hence avoiding fatal instabilities in the numerical integration due to the increased velocities of the atoms in high-temperature replicas. We launched HREM simulations using 4 or 8 replicas with a scaling protocol [23] given by
(1)$$s_{m}=S^{m-1}/\left(N_{\mathrm{rep}}-1\right)$$
where *S* is the minimum scaling factor and $N_{rep}$ is the number of replica. The total length of the HREM sampling on the target state was of 8 ns. All HREM simulations were completed on a local 8-cores work-station in few minutes.

**Solvated APA ST-HREM:** For APA in solution, we used a solute tempering scheme where *only the APA degrees of freedom are scaled* up to 0.1 or 0.05. Hence, NPT condition can be maintained as the water solvent remains cold during the simulation. At variance with the so-called REST2 scheme [37], solute–solvent interactions were also unscaled. Due to this choice, the HREM setup for the solution can be chosen identical to that of the gas-phase. Scaling solute–solvents, by involving the solvent degrees of freedom, would force the use of a high number of replica to allow a good overlap of the energy distributions of contiguous replica. We recall that the number of needed replicas in a HREM simulation grows with $N^{1/2}$, where *N* is the number of degrees of freedom involved in the scaling [23].

**λ-hopping simulations**: HREM simulation with λ-hopping for solvated APA is emulated with ORAC *by scaling only the solute–solvent interaction potential* up to a factor of λ=0.05. This variant of the HREM is used by practitioners in alchemical applications to compute, e.g., the hydration free energy via FEP, assuming that λ-hopping will enhance the sampling. In the λ-space, the potential energy of a λ-state is hence given by $V\left(\lambda \right)=V_{s}+V_{S}+\lambda V_{sS}$, where $V_{s},\;V_{S}$ and $V_{sS}$ are the solvent, solute and solute–solvent potential, respectively. The target state of this HREM variant corresponds to the fully hydrated APA molecule (λ=1), while the state at the smallest λ is characterized by a nearly a decoupled (gas–phase) ligand. Note that such a scheme is equivalent to a ST-HREM where only the solute–solvent interactions are scaled down to 0.05, corresponding to a “solvent-solute temperature” of 6000 K. We used 16 replicas in the range λ=[0.05,1], with a scaling protocol given by Equation (Equation 1). Due to the scaling factor involving the solvent, the number of replicas had to be increased up to 16. The simulations were carried out on the CRESCO cluster [38]. HREM simulations were done starting from an initial configuration of the APA molecule in the cis or trans form.

**λ-hopping with the hot zone (FEP${}^{+}$ emulation)** In this case, the λ-hopping calculation was replicated by adding an *intramolecular* scaling for the solute, i.e., defining the APA molecule as the “hot zone” of the system as in FEP${}^{+}$ for binding free energy calculations using alchemy [39]. In first instance, the intramolecular scaling (including the solute bonded and non-bonded interactions) was chosen to be similar to that proposed in the original FEP${}^{+}$ paper (and possibly in the Desmond [40] default [21] for FEP${}^{+}$) with a minimum scaling factor at intermediate λ of S=0.25, corresponding to a solute temperature of 1200 K) and no intrasolute scaling (S=1) at full coupling λ=1 and the quasi-gaseous state λ=0.05. FEP${}^{+}$ simulations were also performed by scaling the intrasolute (hot zone) potential up to a minimum scaling factor of 0.05, as we did for the standard HREM simulations of solvated APA. For both these two FEP${}^{+}$ simulations, we used in all cases 16 replicas starting from the E or Z state, running on CRESCO from a minimum of 8 ns to a maximum of 32 ns per λ state.

## 3. Results and Discussion

Here we are interested in assessing whether the λ-hopping approach in FEP calculations is capable of achieving the sampling efficiency of an ordinary HREM simulation in solution in cases where conformationally restricted metastable states are present. To this end, we use as metrics the free energy difference of the E/Z ratio, $\Delta G_{E/Z}=-RT\log\left(P_{E}/P_{Z}\right)$, in the *target state* of APA. In Table 1 we collect the results for $\Delta G_{E/Z}$ in the gas-phase and in solution for various HREM protocol and for the HREM-based FEP or FEP${}^{+}$ emulation. Well-behavior of temperature Replica Exchange (t-REM) and HREM simulations is normally assessed by monitoring the round-trip time (RTT) and the exchange ratio (ER). Significant ERs are expected in a well-working REM simulation, as ER is a direct measure [23] of the overlap between the energy distributions regulating the exchange. Short RTTs implies that the replica walkers can easily diffuse with no bottlenecks along the whole thermodynamic range.

### 3.1. t-REM and ST-HREM of APA in the Gas-Phase and in Solution

As can be seen in the first three rows of Table 1, in the gas-phase, $\Delta G_{E/Z}$ is negative, i.e., the Z(cis) state is favored, and can be reliably recovered in just 8 ns with differences of less than 0.5 kcal/mol, no matter the REM protocol, provided the exchange ratio is different from zero for each step of the replica ladder *and* that the scaling factor S is chosen such that barriers can be overcome for $m=N_{\mathrm{rep}}$ in Equation (Equation 1). To this end, minimum scaling factors such that S≤0.1 (corresponding to a 10-fold reduction of the C=C torsional barrier) are necessary. We see from the table that by reducing the number of replica, both the exchange ratio and the round-time decrease. The latter is reduced to 80 ps due to the accelerated diffusion of the four replica walkers in the temperature space, notwithstanding the reduction in the exchange rate (8–44%).

As shown in the central five rows of Table 1, the presence of the solvent has a remarkable impact on the E/Z ratio yielding a positive free energy, hence slightly favoring is solution the E conformer. Note that due to the solvent collisions, the RTT is strongly reduced in solution with respect to the gas-phase. Again, so long that the exchange ratio is different from zero everywhere on the replica ladder, $\Delta G_{E/Z}$ is remarkably independent on the chosen ST-HREM protocol (*S* or $N_{\mathrm{rep}}$ combinations) and on the simulation time span in the range 8–32 ns.

In Figure 2, we show, as a representative example, the t-REM and ST-HREM sampling of the zwitterionic APA molecule in the gas phase (a,b) and in solution (c,d), respectively using in both cases S = 0.1 (corresponding to a temperature of 3000 K) and $N_{\mathrm{rep}}=8$. Despite such high temperature range, as only the degrees of freedom of the solute are involved in the scaling, 8 replicas are amply sufficient for obtaining a good exchange rate (see Table 1). Quite reasonably, the Z (cis) state is strongly favored in the target state of the gas-phase (black circles in Figure 2a) where the attractive interactions between the charged end groups are unscreened. The trans state, on the other hand, while easily and uniformly sampled in the hottest GE state (red circles in Figure 2b), is very rarely transmitted to the target state. In Figure 2b, the gas-phase distribution of the dihedral angle around the double bond is shown for the target state (black) and the hottest thermodynamic state corresponding to 3000 K (red).

The cis/trans exchange in water (Figure 2c,d) is frequent and homogeneous on the target state (black symbols) and on the hottest state (red circles) as well. The trans(E) state in water is now favored in the target state (black curve in Figure 2d), although the Z(cis) state population remains significant. The enhancement of the E(trans) population in water-solvated APA is expected due to the solvent screening of the opposite charge end-groups. The cis-trans population ratio increases in the hottest state (d).

The histograms of Figure 2b,d, referring to the target state (black symbols), allows us to straightforwardly compute the E/Z ratio of APA and the associated free energy change $\Delta G_{E/Z}$ at 300 K, but they give no information on the height of the free energy barriers. Configurations corresponding to the top of the barrier are sampled only at the highest temperature (red symbols) where the barrier is lowered by the *S* factor. In an HREM simulation, however, one can take advantage of the full statistics to obtain equilibrium averages of a given configurational property for any target distribution using the so-called multiple Bennett acceptance ratio (MBAR) [41,42,43]. MBAR weights of all HREM-sampled states can be calculated for the target state distribution at 300 K, hence reconstructing, at this temperature, the full free energy profile along the dihedral angle. The calculation of the MBAR weights is performed with the algorithm described in Ref. [43], using the code mbar provided as supplementary information in Ref. [43] and in the Zenodo repository. In Figure 3, we show the MBAR-determined free energy profile of the dihedral angle around the double bond in APA. To reduce the noise on the top of the barrier, the calculations are done using the scaling *S* = 0.05. With such a scaling, one has a quasi-free rotation around the C=C bond. The stabilization free energy of theE(trans) state of APA in going from the gas-phase to the solution is of approximately 4 kcal/mol. Remarkably, while solvent collisions allow a much more rapid replica diffusion (compare gas-phase and solution RTT’s in Table 1), the barrier height in going from the Z to E state is only slightly affected by the presence of the solvent. This unresponsiveness of the torsional barrier to solute–solvent interactions should indeed raise a red flag for the λ-hopping effectiveness in FEP calculations.

### 3.2. λ-Hopping and FEP${}^{+}$ Results

In the last 6 rows of Table 1, we show the results obtained with λ-hopping for the E/Z ratio. The conformation of the APA target state (λ=1) remains that of the selected initial configuration. In Figure 4a,b, the λ-hopping simulation ($N_{\mathrm{rep}}=16$, 1≤λ≤0.05) was started from the E state and in (c,d) from the Z state. Despite the fact that standard diagnostics (see RTT and ER in Table 1) seems indicative of a well-behaving REM simulation, the pure λ-hopping protocol is ostensibly unable to surmount the Z-E barrier in any of the λ-states. The same state (E or Z) is found in all walkers depending on the starting configuration of the simulation. Therefore, pure *λ-hopping is not enhancing in any way the sampling* in 5-Aminopent-3-enoic acid.

No enhanced sampling of the dihedral angle around the C=C bond is detected, even using FEP${}^{+}$ emulation with a minimum scaling factor for intrasolute interactions of 0.25 at intermediate λ, corresponding to an absolute temperature of 1200 K of the so-called [39] hot-region. Again, with this FEP${}^{+}$ protocol, corresponding to the suggested protocol in Ref. [39] for FEP binding free energy calculations in ligand–receptor systems, the APA molecule remains in the initial configuration during the whole target state sampling (S=1 and λ=1), despite all REM indicators behave seemingly well (see Table 1). The E/Z transition can only be observed using a FEP${}^{+}$ protocol with an intrasolute scaling factor down to S≤0.1, i.e., *using the same scaling factors for a well-converged t-REM or H-REM simulation*. This reduced *S* factor summing up to the λ-scaling at constant $N_{\mathrm{rep}}=16$, increases the round-trip time by more than 50%, hence slowing convergence in FEP${}^{+}$. As shown in Table 1, $\Delta G_{E/Z}$ apparently stabilizes for simulation time of 32 ns at a value (≃0.7 kcal/mol) that is still slightly higher than that observed in the standard ST-HREM simulation (≃0.5 kcal/mol).

### 3.3. Hydration Free Energy of APA with λ-Hopping

In λ-hopping/FEP${}^{+}$ simulations, MBAR allows us to straightforwardly compute the hydration free energy of APA with Z/E equilibrium. MBAR weights are evaluated using the Crooks theorem for instantaneous switches [43] from the ratio of the partition functions $Z_{\lambda }$ of neighboring λ-states. As $Z_{\lambda_{k+1}}/Z_{\lambda_{k}}=e^{\Delta F_{k}}$, with $\Delta F_{k}$ being the free energy difference (in RT units) in going from state $\lambda_{k}$ to state $\lambda_{k+1}$, the free energy connecting the end-states (namely λ=1 and λ=0.05), corresponding to the negative of the hydration free energy (ΔG), can be readily computed as
(2)$$\Delta F=\sum_{k=1}^{N_{\mathrm{rep}}-1}\ln Z_{\lambda_{k+1}}/Z_{\lambda_{k}}$$

The hydration free energies of the E and Z isomers, $\Delta G_{E}$, $\Delta G_{Z}$, are related to ΔG of the thermodynamic mixture as
(3)$$\Delta G=\Delta G_{E}-RT\ln\left(\frac{1}{1+R_{Z/E}}+\frac{e^{-\beta (\Delta G_{z}-\Delta G_{E})}}{1+R_{E/Z}}\right)$$
with $R_{Z/E}$ being the equilibrium Z/E ratio of APA in an ideal solution. We have seen that a simple λ-hopping simulation (no hot-zone) affords the hydration free energy of either the Z or E conformer, depending on the starting configuration of the solvated APA molecule. The solution/gas equilibrium for the Z and E isomers in ideal conditions is given by $K_{E}=c_{E}^{\mathrm{gas}}/c_{E}^{\mathrm{sol}}=e^{-\beta \Delta F_{E}}$ and $K_{Z}=c_{Z}^{\mathrm{gas}}/c_{Z}^{\mathrm{sol}}=e^{-\beta \Delta F_{Z}}$. For the mixture we have $K=\left(c_{E}^{\mathrm{gas}}+c_{Z}^{\mathrm{gas}}\right)/\left(c_{E}^{\mathrm{sol}}+c_{Z}^{\mathrm{sol}}\right)=e^{-\beta \Delta F}$. Using this three relations, after some trivial algebra we arrive at Equation (Equation 3). Equation (Equation 3) can hence be effectively used to verify the accuracy of the FEP^+^ calculation. In Figure 5.

We show $\Delta F_{k}$ along the alchemical stratification computed with pure λ-hopping for the two isomers, and with FEP${}^{+}$ for the equilibrium mixture. We can see that the FEP${}^{+}$ trend is markedly different from that of the pure λ-hopping for the two isomers. The irregular behavior of $\Delta F_{k}$ with FEP${}^{+}$ is due to the superimposition of the *two* scaling factors, λ, referring to the APA-environment interaction, and *S*, referring to the hot-zone involving the APA intramolecular potential.

In Table 2, we finally show the results obtained for ΔG computed with FEP^+^ using a hot-zone scaling factor S=0.05 (as in Figure 5) with 16 replicas and various simulation lengths, compared to Equation (Equation 3) where $\Delta G_{E}$ and $\Delta G_{Z}$ are computed using standard λ-hopping on the E and Z isomers, respectively.

While the free energy difference between the Z and E isomers computed with simple λ-hopping is nearly of 3 kcal/mol, ΔG with FEP^+^ differs by less than 0.2 kcal/mol from the value computed using Equation (Equation 3), showing a remarkable consistency between simple λ-hopping on the individual Z and E species and λ-hopping of the equilibrium mixture with the hot-zone.

## 4. Conclusions

We have thoroughly tested the effectiveness of enhanced sampling techniques in FEP calculations for a paradigmatic example; the 5-Aminopent-3-enoic-acid with central double bonds defining conformationally restricted E(trans) and Z(cis) state, separated by large energy barriers of tens of kcal/mol. High-torsional barriers of this size are commonplace in proteins as well as in ligands, determined by the presence of out-of-ring sp${}_{2}$ bonds or by sterically restrained interconversion of rotamers. As shown by standard Hamiltonian Replica Exchange simulations, the zwitterionic form of the APA molecule undergoes an inversion of stability of the E and Z form in going from the gas-phase to the solution, with the Z form being strongly destabilized by the dielectric screening of the electrostatic interactions between the amino and carboxylic moieties in the solvent. At variance with standard ST-HREM for solvated APA, pure λ-hopping simulations are unable to enhance the sampling of the Z and E isomers anywhere along the λ-stratification, including the fully coupled (target) state. Sampling using the λ-hopping is found to be pathologically dependent on the initial configuration of the APA molecule, consistently remaining stuck in one of the two isomers.

Enhanced sampling can be effectively induced in λ-hopping using a FEP${}^{+}$ protocol where the so-called hot region [39] is restricted to the APA molecule. However, the inattentive use of the FEP${}^{+}$ default scaling factors for the hot region can either prevent the correct sampling of the isomeric states, or can significantly slow down the convergence. As the barrier height might be modulated by the solute–solvent coupling along the λ stratification, the optimal FEP${}^{+}$ minimum scaling factor *S* for the hot region should hence be preliminary tuned by running two ordinary solute-tempering HREM simulations of the end-states at λ=1 (full ligand coupling) and λ=0 (ligand in the gas-phase).

## Figures and Tables

**Figure 1 molecules-27-04426-f001:**
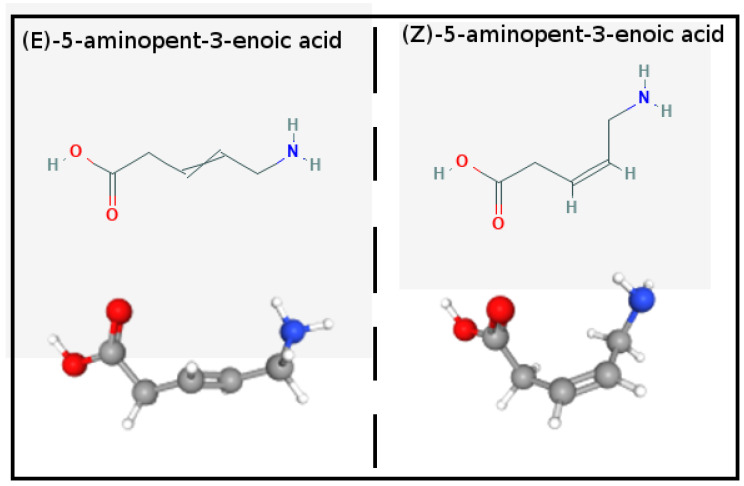
Structure of the E- and Z- 5-Aminopent-3-enoic-acid diastereoisomers.

**Figure 2 molecules-27-04426-f002:**
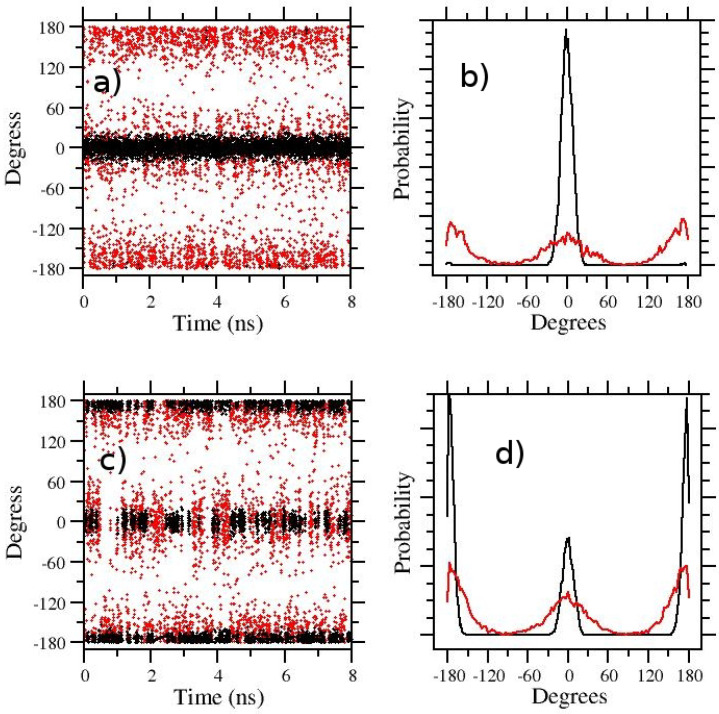
HREM sampling of the 5-Aminopent-3-enoic acid in the gas phase (**a**,**b**) and in OPC3 water (**c**,**d**). In (**a**,**c**) the time records of the central dihedral angle around the double bond are reported for the target state (black circles) and for the hottest state (red circles). The corresponding distribution of the dihedrals are shown in (**b**,**d**). In all cases, the REM protocol is the same (S=0.1, $N_{\mathrm{rep}}=8$).

**Figure 3 molecules-27-04426-f003:**
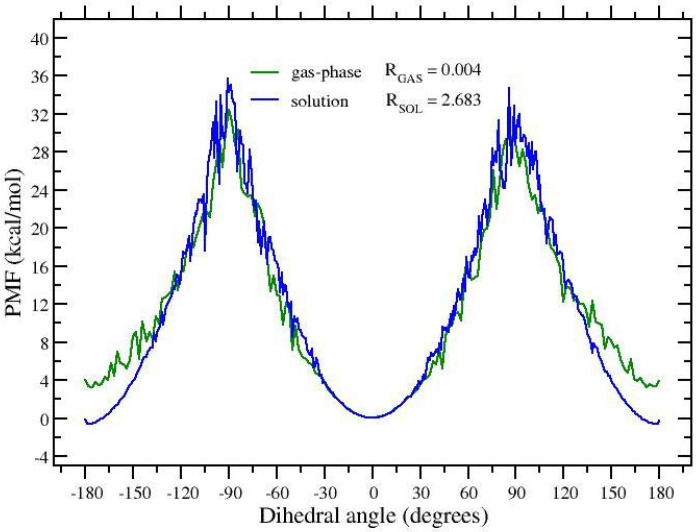
Free energy profile of the dihedral angle in gaseous and solvated APA at 300 K reconstructed with MBAR from HREM simulations done with the protocol S=0.05, $N_{\mathrm{rep}}=8$.

**Figure 4 molecules-27-04426-f004:**
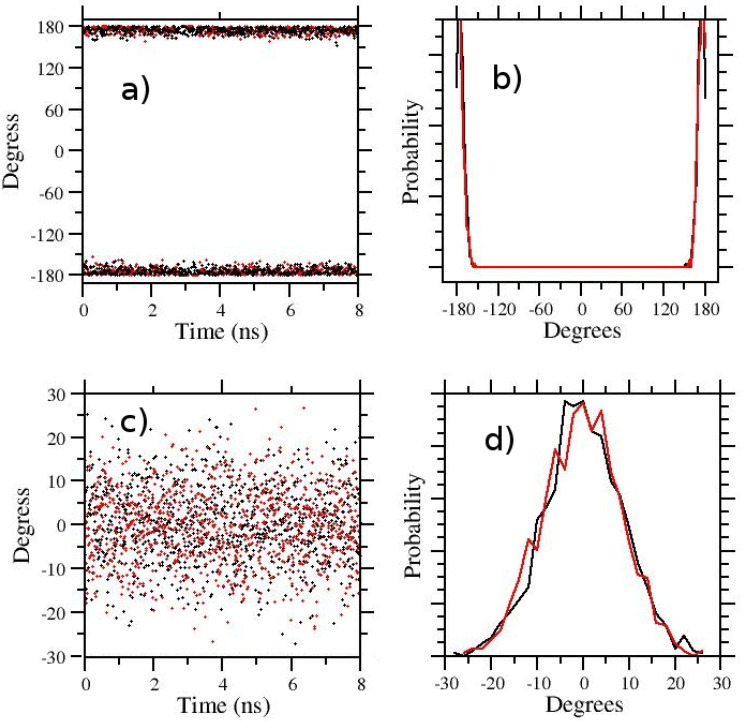
HREM sampling of the 5-Aminopent-3-enoic acid in TIP3P water with HREX/lambda-hopping. In (**a**–**d**) the starting configuration were the trans and cis state, respectively. Time records are on (**a**,**c**) and dihedral distributions on (**b**,**d**). Black and red symbols/line in ab,cd refer to the λ=1 and λ=0.1 states.

**Figure 5 molecules-27-04426-f005:**
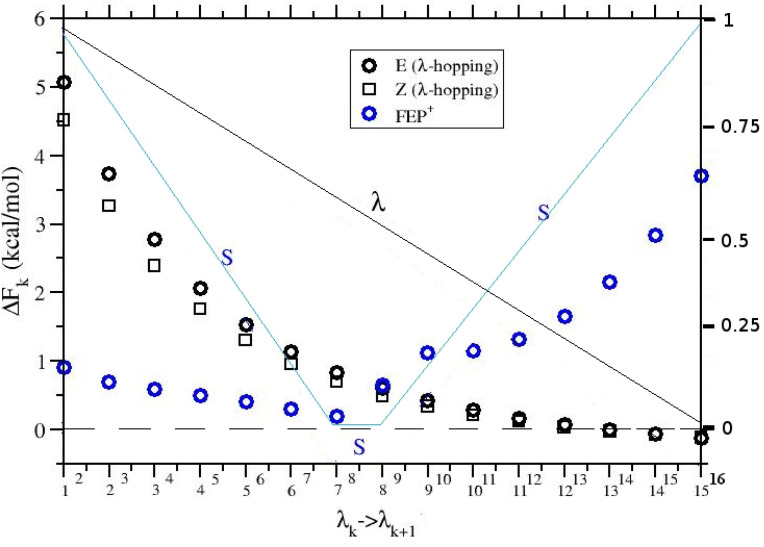
$\Delta F_{k}$ for the $\lambda_{k}\to \lambda_{k+1}$ transitions (black symbols) in λ-hopping for the E and Z isomers and in FEP${}^{+}$ (blue symbol). The right y-axis refers to the λ (black line) and *S* (blue line) scaling factors.

**Table 1 molecules-27-04426-t001:** Round-trip times (RTT) and Z/E free energy difference of the target state of APA for t-REM in the gas-phase, ST-HREM in solution, λ-hopping and FEP${}^{+}$ in solution a function of the enhanced sampling protocol (Rep.—number of replicas; Time/ps—simulation time per replica; Exch—acceptance exchange ratio). N.B.—in all simulations stretching and bending are unscaled.

Gas-Phase						
	Rep.	S	Time/ns	Exch.	RTT/ps	$\Delta G_{E/Z}$
t-REM	8	0.1	8.0	42–58%	103 ± 12	−3.0 ± 0.5
t-REM	8	0.05	8.0	15–44%	134 ± 14	−2.8 ± 0.6
t-REM	4	0.05	8.0	8–44%	80 ± 3	−3.3 ± 0.8
Solution						
	Rep.	S	Time/ns	Exch.	RTT/ps	$\Delta G_{E/Z}$
ST-HREM	8	0.1	8.0	58–81%	6.9 ± 0.7	−0.17 ± 0.07
ST-HREM	8	0.05	8.0	44–79%	7.1 ± 0.6	0.63 ± 0.1
ST-HREM	8	0.05	16.0	44–79%	7.1 ± 0.6	0.53 ± 0.06
ST-HREM	4	0.05	16.0	15–44%	6.3 ± 0.4	0.59 ± 0.06
ST-HREM	4	0.05	32.0	15–44%	6.3 ± 0.3	0.51 ± 0.05
Solution (FEP/FEP${}^{+}$ with λ-hopping)						
	Rep.	S	Time/ns	Exch.	RTT/ps	$\Delta G_{E/Z}$
λ-hop	16	1.0	12.0	75–87%	11 ± 1	n/a
λ-hop ${}^{+}$	16	0.25	12.0	75–87%	15 ± 1	n/a
λ-hop ${}^{+}$	16	0.1	12.0	47–84%	15 ± 2	0.81 ± 0.22
λ-hop ${}^{+}$	16	0.05	8.0	33–78%	24 ± 2	0.70 ± 0.11
λ-hop ${}^{+}$	16	0.05	16.0	33–78%	24 ± 2	0.68 ± 0.09
λ-hop ${}^{+}$	16	0.05	32.0	33–78%	24 ± 2	0.75 ± 0.07

**Table 2 molecules-27-04426-t002:** Convergence of hydration free energies (in kcal/mol) of APA in standard condition. The value in parenthesis have been obtained using the $R_{Z/E}$ ratios of the target state from the ST-HREM simulation (see Table 1).

Time/ns	$\Delta G_{E}$	$\Delta G_{Z}$	ΔG (Equation (Equation 3))	ΔG (FEP${}^{+}$)
8	−18.61	−15.97	−18.44(−18.43)	−18.29
16	−18.57	−15.91	−18.41(−18.36)	−18.25
32	−18.58	−15.92	−18.41(−18.35)	−18.24

## Data Availability

All data to reproduce the results presented in this study can be downloaded from the general-purpose open-access repository Zenodo (Available online: https://zenodo.org/record/6665606) (accessed on 20 June 2022). The Zenodo archive includes the ORAC [36] input files and technical details for running the HREM/λ-hopping/FEP${}^{+}$ simulations and the PrimaDORAC [29]-generated potential parameters for APA. In the repository we also provide ancillary scripts (with essential documentation) to compute (i) the round-trip time, (ii) exchange ratio via the energy distribution overlap long the replica progression, (iii) the APA cis–trans ratio in all replica states and (iv) the free energy difference between the end-states using MBAR and the hydration free energy. The ORAC program (v6.1) is available for download under the GPL at the website http://www1.chim.unifi.it/orac/ (accessed on 20 June 2022), The mbar program for MBAR calculations [43] is provided in the Zenodo repository.

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
