# Peer review of "Does Hamiltonian Replica Exchange via Lambda-Hopping Enhance the Sampling in Alchemical Free Energy Calculations?"

_molecules, 2022, doi:10.3390/molecules27144426_

Round 1

Reviewer 1 Report

The manuscript submitted by Prof. Procacci describes an analysis of new numerical methods to improve sampling in free energy perturbation (FEP) calculations. The author used the compound 5-amino-3-enoic acid (APA) as a case and explored the E/Z equilibrium. Although the case study is very specific, I think the discussion is interesting and can add to the broad field of FEP in general. Therefore, I would like to recommend the manuscript for publication after minor review considering the points discussed below.

1) Just before Fig. 1, the pKa was given for the carboxylic and amino groups: 4.49 and 9.65, respectively. At pH 7, the molecule must be predominantly as a zwitterion; however, Fig. 1 showed the molecules in the ‘neutral’ form. Please make clear which form was simulated.

2) Experimental data are not available, but authors could do QM calculations or search for theoretical studies in the literature. These QM data would support the results obtained.

3) Fig. 2 is not clear which structure was used in the beginning of simulation. I understand that for Z, the dihedral is 0 degrees and for E it is +/- 180 degrees; however, in Fig. 2c there are some black dots (target state) at 0 and +/- 180?

4) I could not visualize the statement “…The enhancement of the E(trans) population in water-solvated APA is expected due to the solvent screening of the opposite charge end-groups…” (p. 6, line 198). In both forms the end-groups are charged and is expected to interact with water. The RDF could be analyzed to show the differences in the solvation of both forms.

5) p. 7, the statement “…Configurations corresponding to the top of the barrier are sampled only at the highest temperature…” could be verified by counting for frames with the dihedral +/- 90 degrees. However, looking at the histograms in Fig. 2 the number of frames with dihedral = +/- 90 was zero. Could you comment about this?

6) Fig. 3 shows a barrier of 36 kcal/mol. Could you support this value? I thought it too small for a rotation around a double-bond.

Author Response

1) Figure 1 has been generated by PUBCHEM. In the
revised version I now make clear that I simulated the zwitterionic form.

2) What we would need here is the experimental free energy
difference between the two species at room temperature. QM yields the
optimized structure and the potential energy at 0 K with or without
dielectric (i.e. with or without Polarizable Continuum Model PCM with
ε=78). For example, a Gaussian calculation of the initial
zwitterionic molecule in the gas phase does preserve the initial Z or
E structure but in both cases, it ends up in the neutral form with the
HN migrating to the oxygen. (see Gaussian log files in the zipped archive QM.zip at the link https://drive.google.com/file/d/1wmbMxHq8e8ctsnFjQmFe7cNipK_uL4b9/view?usp=sharing)

3) The starting structure (Z or E) is irrelevant in this case. In a
well functioning HREM, as figure 2 shows, there is a continuous
exchange between Z and E isoforms

4) The Z form is favored in the gas phase, as in this conformation
the two groups with opposite charges can get closer with respect to
the E form. In solution, both charged groups can interact
effectively with water. This disfavors to some extent the formation
of the cis state with respect to the trans state.

5) This is a good point. Actually, in figure 2 at the highest
temperature, there are a few red points around 90 degrees, both in the
gas-phase and in solution. This can be appreciated also in the red
histograms reported in panels b) and d) that are small but nonzero near
θ=90. Figure 2 , as indicated in the caption, refers to an HREM
with S=0.1. With such a scaling factor using MBAR the barrier top
(theta =90 ) could hence still be seen but it would be very noisy due
to the poor sampling at θ=90with S=0.1 as the referee noted.

The plot in Figure 3, where the barrier is reconstructed
using MBAR, was done with S=0.05 where the rotation around the C=C
bond is essentially free. This is now stated in the revised version and
the S=0.05 value is now specified in the caption of Figure 3.

6) According to the tpg files provided in the zenodo repository, the
GAFF2 torsional potential around the C=C bond is 5.29 kcal/mol for
the CC=CC dihedral and 6.65 kcal/mol for the CC=CH dihedral. This
yields a purely torsional barrier of ~ 26 kcal/mol. The barrier is
higher than that due to the 14 nonbonded contributions to the full
torsional energy.

N.B. In the attached file, I provide the Gaussian output files for the
optimized structure of the Z and E form in vacuo and in a dielectric
with eps=78 and a pdf version of the manuscript with changes
highlighted in red color. 

Reviewer 2 Report

A clear, convincing and well-written paper.

A few details, typos, etc:

p1, l19: $5,682.3 -pretty sure that some of these digits are not significant.

p2, l55: HREM is already defined (l47).

p2, l67: HREX is a newcomer.

p2, l80: APA is sometimes 5-amino and sometimes 5-aminopent (both in Figure 1).

p4, l138: "HREM. Simulations" -> HREM simulations

p4, l160: t-REM is a newcomer.

p5, legend of Table 1: "Rond-trip" -> Round-trip

p7, l205: "is lowered of the S factor" -> of the S factor is lowered

p9, l251: two isomerS.

p10, legend of Table 2: missing ). at the end.

p11, l311: "Eree" -> Free. The list of abbreviations could be more extensive.

p13, l406: "Using this three relation" -> Using these three relations

Author Response

I thank the reviewer for his positive comments and for carefully proofreading the manuscript. 
